# Controllable Synthesis of Special Reed-Leaf-Like Carbon Nanostructures Using Copper Containing Catalytic Pyrolysis for High-Performance Field Emission

**Chen Zhao [1], Zhejuan Zhang [1,*], Jun Guo [1], Qiang Hu [1], Zhuo Sun [1] and Xianqing Piao [2]**

[1] Department of Physics, Engineering Research Center for Nanophotonics &Advanced Instrument, Ministry of Education, East China Normal University, North Zhongshan Rd 3663, Shanghai 200062, China; Chem_zhao@hotmail.com (C.Z.); bailaohui@hotmail.com (J.G.); atom_h@163.com (Q.H.); zsun@phy.ecnu.edu.cn (Z.S.)

[2] Shanghai Industrial Technology Institute, Jinsu Rd 200, Shanghai 201206, China; Xqpiao@phy.ecnu.edu.cn

* Correspondence: zjzhang@phy.ecnu.edu.cn; Tel.: +86-021-622-34323

**Abstract:** Special reed-leaf-like carbon nanostructures have been realized by using chemical vapor deposition (CVD) under the combined action of copper containing catalytic pyrolysis and ammonia ($NH_3$) gas. The nucleation and growth mechanisms of CNLs based on growth parameters are discussed. The Raman spectra of carbon nanotubes (CNTs), CNLs and CNT-CNL composites were measured and found to be strongly influenced by the type of gas. Field emission (FE) properties of CNL-CNT composites were observed with a lower turn-on electric field of 0.73 V/μm, and a higher current density of 18.0 mA/cm$^2$ at an electric field of 2.65 V/μm, which are superior to those of CNTs and flower-like CNLs. This is because there are more field emitters in CNLs inter-planted in CNTs. We consider that the unique FE stability of CNTs and defects in CNLs play a synergetic role on the improved FE properties.

**Keywords:** carbon nanostructure; copper catalyst; field emission properties; catalytic pyrolysis

## 1. Introduction

Carbon-based materials fabricated by catalytic chemical vapor deposition (CVD), such as graphene, carbon nanotubes (CNTs), carbon nanofibers (CNFs), diamond and microcrystalline graphite, have good electrical properties [1–4]. They are considered as ideal candidates for field emitters [5,6]. Especially, CNTs exhibit superior structural advantages over other allotropes in terms of emitting electrons due to their unique sharp emitting nano-tips that increase local field strength easily [7,8].

Recently, carbon nanocomposites have been attracting increasing attention because they can integrate the advantages of various carbon materials, thus obtaining any desired properties and broadening their application field. Recent works have focused on the synthesis of graphene-CNT on a copper catalyst for high-performance field emission (FE), and some progress has been made. Deng et al. [9] introduced the fabrication of graphene-vertical CNTs (GF-CNT) with higher FE stability. However, the GF-CNT shows poor FE properties due to the shield of field enhancement from CNTs, which limits the improvement of the FE current density [10]. Furthermore, our previous work on carbon nanostructures (CNS) has shown that Cu-catalyzed coiled CNS showed an optimized FE property, compared with that of straight CNTs [11]. Therefore, it is believed that the FE performance of CNT field emitters can be further improved by combining the advantages of CNS in special shapes.

This paper introduces a simple approach to synthesize carbon composites through a copper containing catalytic pyrolysis method. The growth of these reed-leaf-like carbon nanostructures (CNLs) using chemical vapor deposition (CVD) is achieved by controlling the catalyst and gas. We studied the internal relations between the catalysts and reaction carrier gas. Additionally, the effects of surface morphologies of CNS on the performance of FE were also investigated.

## 2. Experimental

Copper catalyst films with thicknesses of 100 nm and 500 nm were deposited on glass substrates using a DC magnetron sputtering method, while Cu-Cr (1:1) catalyst films of the same thicknesses were deposited using co-sputtering. The thicknesses of the thin films were measured by a surface profiler (Detak 6M, Digital Instruments/Vecco Metrology, Santa Barbara, US). The substrates coated by Cu or Cu-Cr alloy films were placed in a 10″-diameter quartz tube furnace at room temperature and the pressure of the furnace was reduced to less than $2 \times 10^{-2}$ Pa using a mechanical pump. Firstly, the substrates were preheated at 650 °C for 5 min in a reaction carrier gas, such as hydrogen ($H_2$) or ammonia ($NH_3$), with a gas flow rate of 60 sccm. Then, the CNS were subsequently grown in the mixture of acetylene (240 sccm $C_2H_2$) and reaction carrier gas (60 sccm). The temperature of the substrate and the pressure of the chamber were kept at 650 °C and $1 \times 10^2$ Pa for 60 min. At last, the samples were furnace-cooled in a nitrogen flow environment.

The surface morphologies of the catalysts and the synthesized CNS were observed by atomic force microscope (AFM, Digital Instruments/Vecco Metrology, Santa Barbara, US, America), scanning electron microscopy (SEM, JSM-LV-5910, JEOL, Tokyo, Japan) and transmission electron microscopy (TEM, Hitachi H-600, Hitachi, Tokyo, Japan). A Raman spectroscopy system (Renishaw micro-Raman System, Renishaw, London, England) with a wavelength of 514 nm was used to identify the structure and the crystalline structure of carbon products. The FE properties of the samples were studied by a diode configuration that had been described in our previous study [12]. The anode was made from indium tin oxide coated glass and the thickness of the insulating spacers between the anode and cathode were 170 μm.

## 3. Results and Discussion

### 3.1. Morphologies and Structures

Figure 1 shows the AFM images of different catalyst films after annealing at 650 °C in a vacuum for 60 min. The morphologies of the Cu films in Figure 1a,b are different from those of the Cu-Cr films in Figure 1c,d. After a high temperature treatment, the surface of the film was granular. Meanwhile, the size and surface roughness of the catalyst film increased with the thickness of the film. For the film with larger thickness, a longer deposition time during the magnetron sputtering process results in the increment in substrate temperature and nucleation rate, which leads to a larger particle size on the surface of the catalyst film [13]. Further, the melting point of the alloy is generally lower than that of the single metal, so the particle size and surface roughness of the Cu-Cr catalyst film is bigger than that of the Cu catalyst film, as shown in Figure 1c,d, respectively.

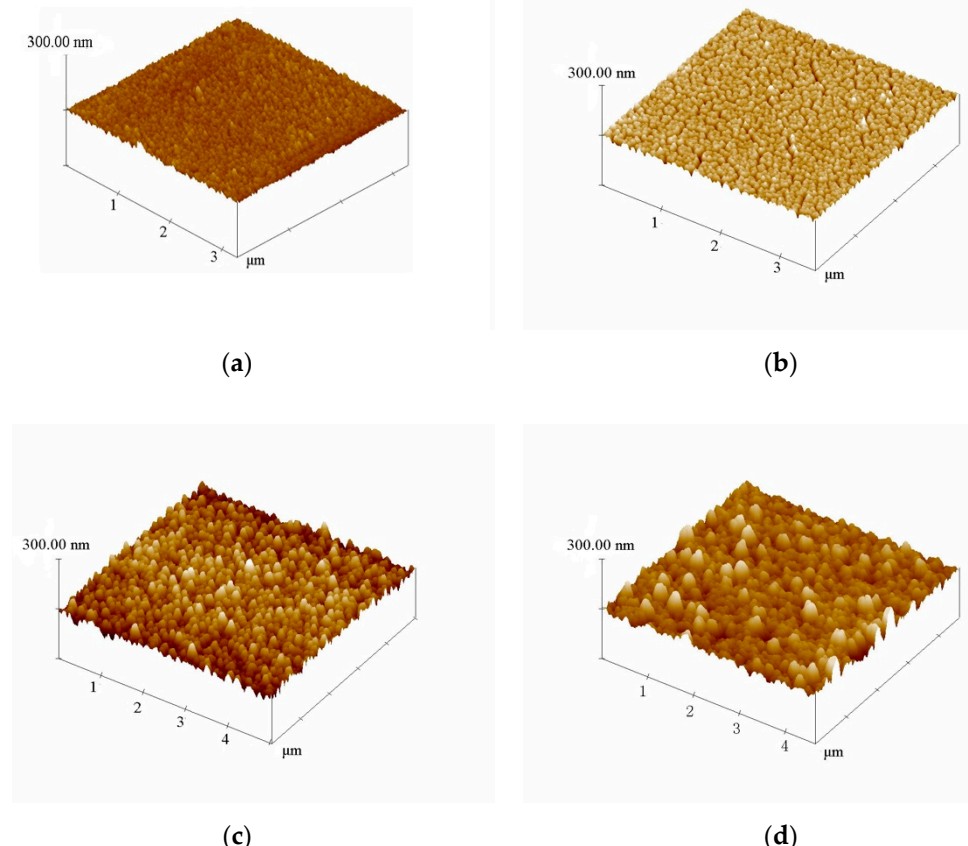

**Figure 1.** AFM images of different catalysts annealing at 650 °C for 60 min. (**a**) 100 nm Cu film; (**b**) 500 nm Cu film; (**c**) 100 nm Cu-Cr alloy film; (**d**) 500 nm Cu-Cr alloy film. AFM: Atomic Force Microscope.

Figure 2 shows SEM images of different CNSs after $C_2H_2/H_2$-CVD, where the inserts are the SEM images of the corresponding catalyst films annealed in a $H_2$ atmosphere. As shown in Figure 2a,b, CNTs with a diameter of nearly 40 nm dispersed on the Cu catalyst. The density of the CNTs increased with the thickness of the catalyst film. As for the Cu-Cr catalyst film of 100 nm, there were many graphite coated particles mixed in the CNT products, as shown in Figure 2c. Comparing Figure 2c with Figure 2a,b, the density of the CNT products on the Cu-Cr catalyst decreased significantly. When the thickness of the Cu-Cr film increased to 500 nm, in addition to some sub-micron CNFs, the density of the CNTs increased slightly. This difference in morphology can mainly be attributed to the fact that the size of surface particles in the Cu catalyst film are smaller than in the Cu-Cr film, as shown in the inserts, resulting in the faster growth rate of CNTs during the CVD process. It is known that the bonding of metal with carbon atoms will increase with the number of unfilled d-orbitals. Cu $(3d^{10}4s)$ has no d-vacancy, which is expected to show minimal affinity with the CNTs. Even so, the low solubility of carbon in Cu is enough for the carbon atoms to form CNTs of small diameters, according to the vapor-liquid-solid (VLS) mechanism [14]. However, catalyst particles of larger sizes are not conducive to the growth of CNTs, so the density of CNTs decreases on Cu-Cr catalysts, as shown in Figure 2c,d. In addition, some lattice distortions are produced when Cr atoms are introduced into Cu crystals and act as interstitial sites after co-sputtering and annealing, which increases the adsorption and deposition rate of carbon atoms on the surface of catalyst particles according to the defect growth mechanism [15]. Half-filled d-orbitals of Cr enhance the affinity with CNTs. As a result, the concentration of carbon atoms dissolved in bigger Cu-Cr particles is so high that sub-micron CNFs are formed.

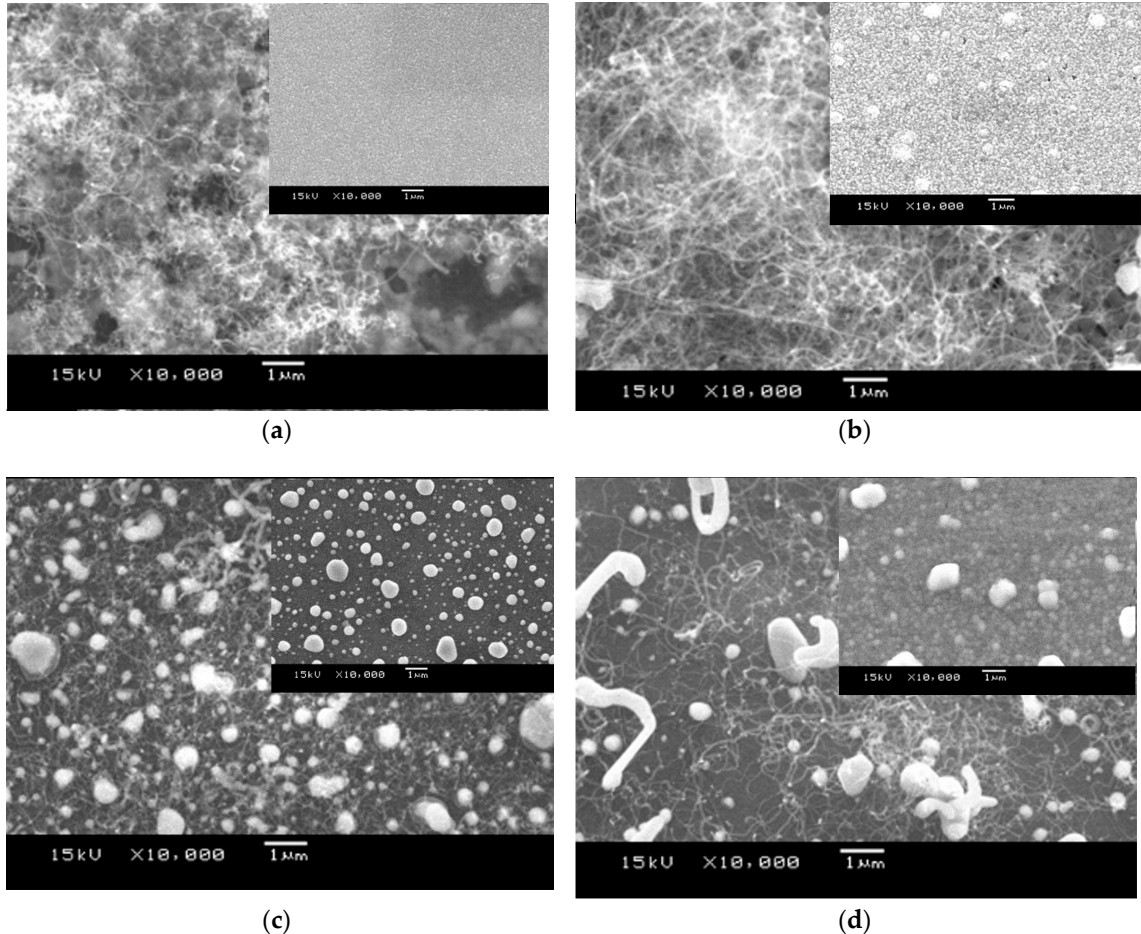

**Figure 2.** SEM images of different carbon nanostructures (CNS) after $C_2H_2/H_2$-chemical vapor deposition (CVD): (**a**) 100 nm Cu film; (**b**) 500 nm Cu film; (**c**) 100 nm Cu-Cr alloy film; (**d**) 500 nm Cu-Cr alloy film.

Previous works [15–21] have reported that ammonia gas has an etching effect and is favorable for the growth of CNTs. In this study, when $H_2$ was replaced by $NH_3$, the morphologies (shown in Figure 3) of carbon products changed greatly. The diameter of CNTs on the Cu film of 100nm increased greatly, and they become shorter and more distorted. Figure 3b is the TEM image of CNTs in Figure 3a, where CNTs with a bamboo-like structure (indicated by circle) can be found (a similar structure is commonly observed in works using $C_2H_2/NH_3$-CVD [2]), indicating a tip-growth mechanism. In Figure 3c, there are many CNLs mixed with CNTs. Figure 3d shows the SEM images of several isolated CNLs from the sample in Figure 3c with higher magnification, in which the folded two-dimensional flake structure of CNL is revealed. The relationship between the size of the catalyst particles and the thickness of the film is consistent with the findings of Juang et al. [22]. Differently, the growth of carbon products is not limited in one mode. It is considered that the mixed structure of CNTs and CNLs may attribute to the mixed growth mode. Tip-growth is more likely to occur on the surface of catalyst particles with a radius below 10 nm, while root-growth tends to appear on particles bigger in size [22]. The size of the catalyst particles on the film of 100 nm were relatively small, so tip-growth was dominant. When the thickness of the Cu film increased to 500 nm, the increment in size results in the appearance of the root-growth mode, leading to the growth of CNL. Inevitably, on the Cu film of 500 nm, bamboo-shaped CNTs based on the tip-growth mode could grow on some small-sized particles synchronously, which is agreement with Figure 3c. To further prove our hypothesis, TEM images of CNL are shown in Figure 3e. The facts, that the top of CNL is narrower than the bottom and the presence of elongated catalyst at the end of CNL (indicated by an arrow and

will be explained in the next paragraph), are strong evidence for the root-growth mechanism of this reed-leaf-like nanostructure.

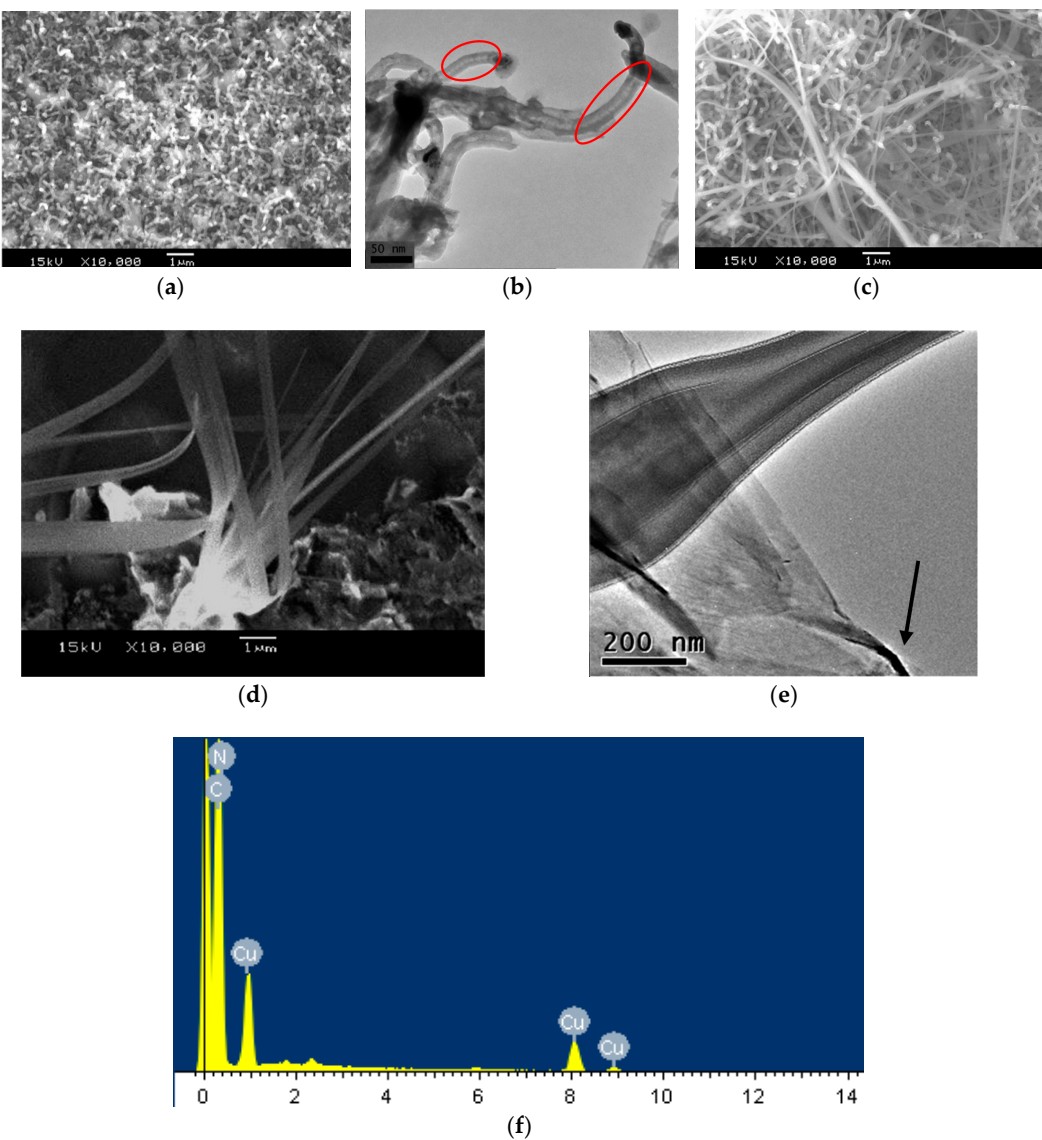

**Figure 3.** SEM (**a**) and TEM (**b**) images of carbon nanotubes (CNTs) grown on 100 nm Cu catalysts in $NH_3$. SEM (**c**), (**d**) and TEM (**e**) images of carbon nanostructures grown on 500 nm Cu catalysts in $NH_3$. EDX(**f**) spectra of carbon nanostructures grown on a 500 nm Cu catalyst in $NH_3$.

A possible root-growth mechanism for CNLs in our thermal system is shown in Figure 4. Firstly, carbon atoms of a certain concentration are supplied to the Cu catalyst due to the adsorption and catalytic decomposition of $C_2H_2$. Then, the carbon atoms diffuse to lower the concentration (through bulk or surface diffusion). At last, precipitation and the formation of the graphite structure can occur. On bigger catalyst particles, the controlling step is the limited carbon diffusion rate. In a $C_2H_2/NH_3$ atmosphere, $NH_3$ inhibits the appearance of amorphous carbon at the initial stage of the synthesis, and thereby protects the Cu metal particles from being covered by amorphous carbon. The activity of carbon can be described as per the following equation [23]:

$$\alpha_C \approx X_C \exp\left(\frac{G_C^0 + \Omega_{CuC} - G_C^{0g}}{RT}\right) \exp\left(\frac{2X_C\Omega_{CuC}}{RT}\right) \exp\left(\frac{X_N W_{NC}}{RT}\right) \tag{1}$$

$X_C$ and $X_N$ are the concentration of carbon and nitrogen atoms in the vapor-liquid-solid (VLS) growth mechanism of CNTs; R and T are the gas constant and temperature, respectively. $G_C^0$ and $G_C^{0g}$ are the Gibbs free energy of the carbon and graphite. $\Omega_{CuC}$ is the interaction between the Cu and C, where $W_{NC}$ is the interaction parameter between N and C at a Cu–N–C resolution. According to [23], $\Omega_{CuC} \leq 0$ and $W_{NC} > 0$. After $H_2$ is replaced by $NH_3$, $X_N \neq 0$. Thus, from Equation (1), it is clear that the activity of carbon is enhanced with the increment of $X_C$ and $X_N$. The graphite layer is formed on the surface of copper particles one by one and the growth rates of the outer graphite layers are faster than the inner ones. When there are one or few layers of graphite appearing on the catalyst surface, the concentration of carbon atoms near the top side will lack precipitation, leading to a bigger void part after the previous graphite layers are lifted up. In addition, the changes in surface energy and elastic energy of the carbon layers distort the surface particles into an irregular shape, as shown in Figure 4b. The curvature at the top of the distorted catalyst particle become smaller, turning Cu tips into spindle-like or even strip-like shapes. Consequently, the concentration of the carbon atoms at the top of Cu particle increases, so more cone-shaped graphite is formed at the surface of the distorted metal particle with a bigger void part. Meanwhile, nitrogen precipitates from the top side of the near-spindle-like metal to the graphite sheet, and some carbon atoms in the graphite sheet are replaced by nitrogen atoms, leading to the distortion of the graphite sheet. The above suggestion is supported by the results in the EDX (Energy Dispersive X-ray Spectroscopy) spectra of carbon nanostructures (as shown in Figure 3f), in which a lot of nitrogen elements have remained.

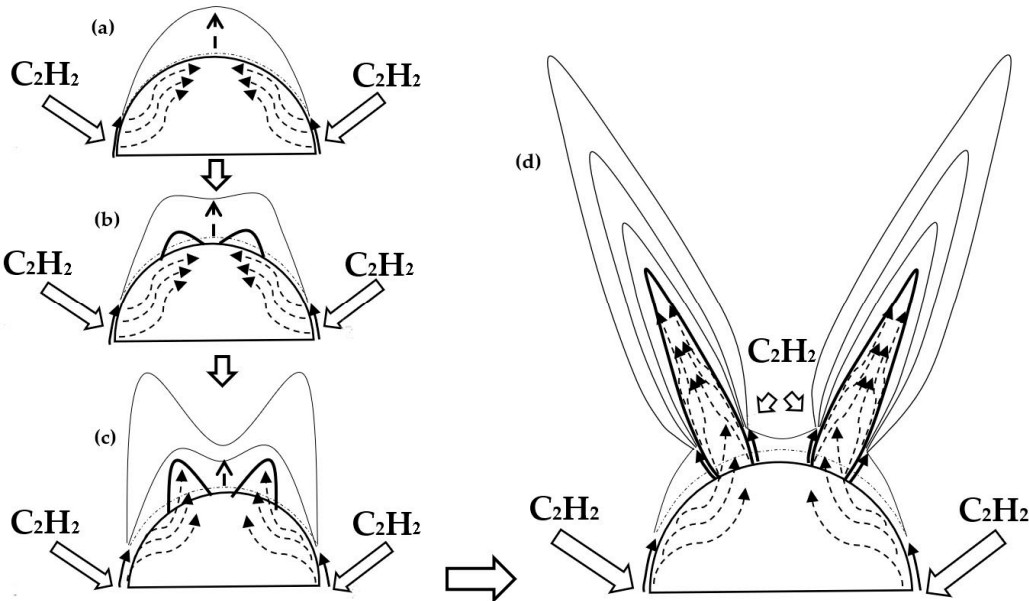

**Figure 4.** The proposed mechanism of the reed-leaf-like carbon nanostructure (CNL).

To find further evidence, the decomposition of CNS on different Cu-Cr catalysts was investigated. Figure 5 shows the SEM images of carbon nanostructures on Cu-Cr films of different thicknesses in a $C_2H_2/NH_3$ atmosphere, where the inserts are the SEM images with low magnification. In Figure 5a, the CNTs that grow on small particles are short and curved, while the big Cu-Cr particles are wrapped over by translucent curved graphite. When the thickness of the Cu-Cr film increases to 500 nm, as shown in Figure 5b, CNLs grow into flowering shrubs on big catalyst particles (approximately 1 μm), which is consistent with the root-growth pattern mentioned above. It is rather surprising to find that the special CNL can be catalytically deposited on the copper-containing catalyst during $NH_3$-CVD.

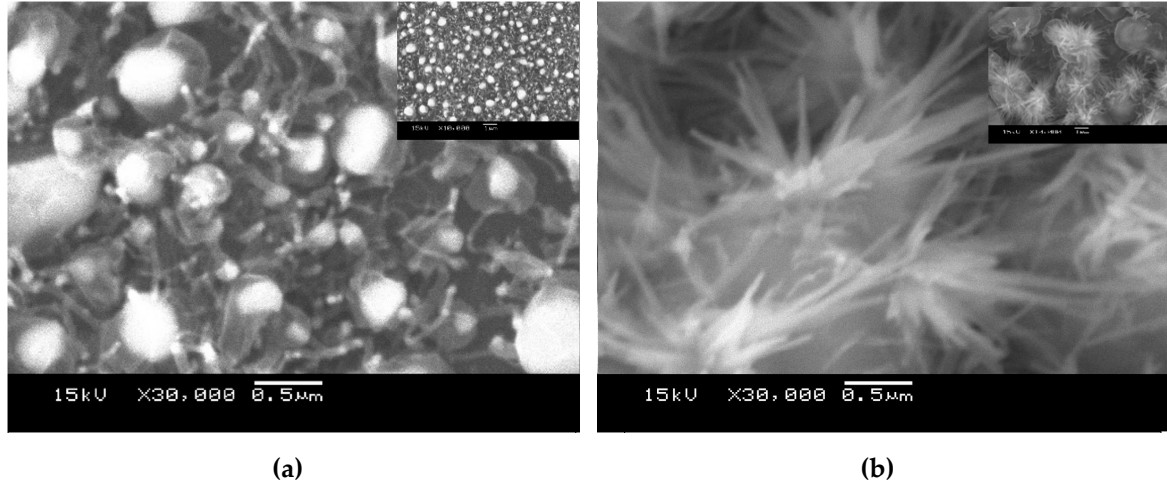

**(a)**                                             **(b)**

**Figure 5.** SEM images of carbon nanostructures on Cu-Cr film of different thicknesses by the CVD method in the presence of $NH_3$ gas. (**a**) 100 nm; (**b**) 500 nm.

### 3.2. Raman Spectroscopy of Carbon Product

The crystalline quality of the carbon nanostructures can be determined by assessing the shape evolution of the Raman spectra of the corresponding samples [24]. The position, width and relative intensity of bands are modified according to the carbon forms [25]. The Raman spectra of different samples are shown in Figure 6. All the spectra of the carbon nanostructures consist of two characteristic peaks, however, the positions of the D and G bands both shift to a higher wavelength, with a decreased intensity of the G band. The Raman bands of Figure 6a are at 1365 $cm^{-1}$ (D band) and 1590 $cm^{-1}$ (G band). The position of the G band in Figure 6c moves to 1677 $cm^{-1}$, revealing the existence of "C=N" bands besides "C=C" bands. It is also consistent with the result of the EDX spectrum in Figure 3f. The ratio of $I_D/I_G$ is derived from the quantitative measurements of the Raman spectra of each sample. The $I_D/I_G$ value for the carbon products on 500 nm Cu and 500 nm Cu-Cr in $NH_3$ were calculated to be 1.2 and 1.32, respectively, suggesting these two types of CNLs are rich in defects.

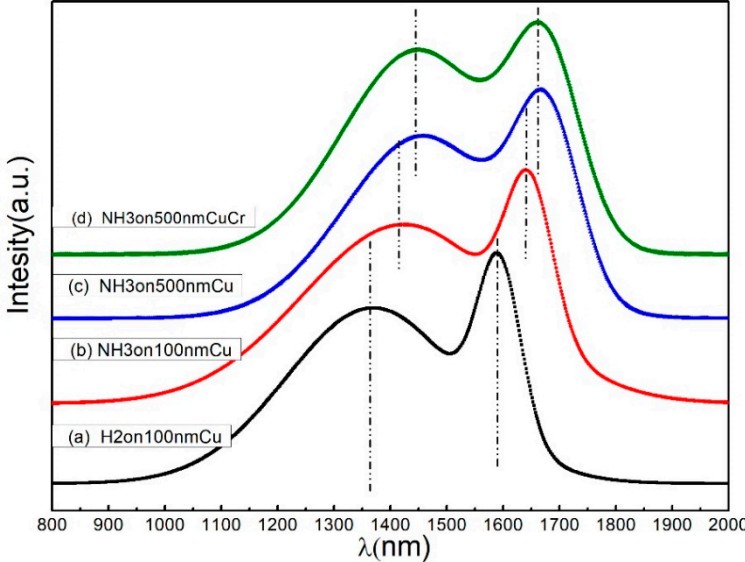

**Figure 6.** Raman spectra of carbon products synthesis using different reducing gases. (**a**) CNTs on 100 nm Cu in $H_2$; (**b**) CNTs on 100 nm Cu in $NH_3$; (**c,d**) carbon products on 100 nm Cu and Cu-Cr in $NH_3$.

### 3.3. Field Emission Properties of Different Carbon Nanostructures

Figure 7 shows the FE performance of the samples. The threshold electric field ($E_{th}$, field at 1 mA/cm$^2$) of the CNT-CNL composite was 0.73 V/μm, which was lower than that of the CNTs (1.32 V/μm). The improved $E_{th}$ can be ascribed to the introduction of defects and wrinkles that increase the number of active emission tips [26]. The electric arc erosion of CNLs occurs easily, and the defects in CNLs are unfavorable to electron transport. Therefore, when E is above 1.68 V/μm, the current density of CNTs and the CNT-CNL mixture increase substantially, followed by CNLs. When E is 2.65 V/μm, the current density of CNT-CNL is higher than that of the other two samples, at 18.0 mA/cm$^2$. The above results indicate that CNL-CNT composites obtained by introducing NH$_3$ into the intermediate process can lead to better FE properties, which is even more pronounced than in vertical graphene-CNT(VG-CNT) composites, as reported by Deng et al [27].

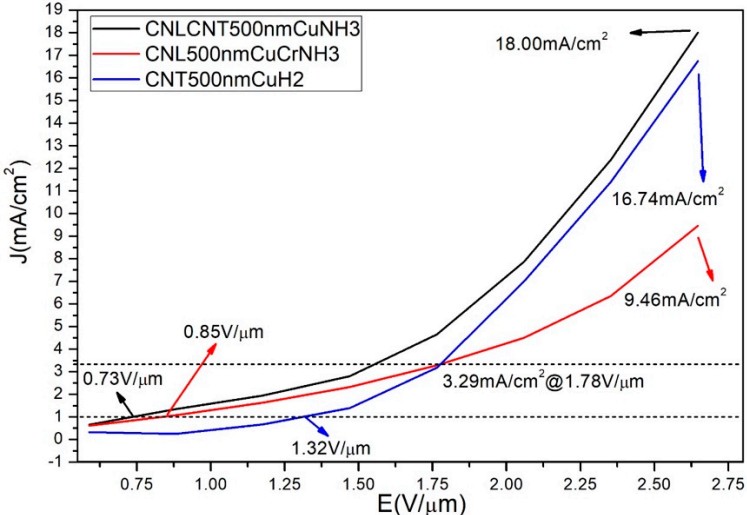

**Figure 7.** Field emission (FE) performance of different CNS.

## 4. Conclusions

Based on the above study of the morphologies and structures of CNLs that were fabricated using a copper-containing catalyst in a NH$_3$/C$_2$H$_2$ atmosphere, the effect of ammonia gas and the catalyst on the preparation process can be confirmed. Copper containing catalyst particles have a faster deposition rate than carbon atoms, and this leads to a higher carbon yield, which contributes greatly to the uniform distribution of carbon nanoproducts on the surface of the catalyst particles. Nitrogen atoms from the reducing gas (NH$_3$) promote the production of CNLs, which complement with the advantages with CNTs. The FE results showed that the threshold electric field of the CNT-CNL composites reduced to 0.73 V/μm as a result of more emission tips appearing due to the defects in the CNLs, while the current density increased from 15.6 mA/cm$^2$ in the CNT to 18.0mA/cm$^2$ at an electric field of 2.65 V/μm. CNLs can act as electron inducers and internal field emitters in the CNT film and improve the FE performance effectively, which could attribute to its' folded, thin shape and defect-rich structure. This work shows that CNL/CNT composites are promising candidates for cold cathode field electron emission.

**Author Contributions:** Conceptualization, C.Z.; methodology, C.Z.; software C.Z.; validation, C.Z., Z.Z., J.G. and Q.H.; formal analysis, C.Z.; investigation, C.Z.; resources, C.Z.; data curation, C.Z.; writing—original draft preparation, C.Z.; writing—review and editing, C.Z., Z.Z.; visualization, C.Z.; supervision, Z.Z.; project administration, Z.Z., Z.S. and X.P.; funding acquisition, Z.Z.

**Funding:** This research received no external funding.

**Acknowledgments:** This work is financially supported by National Nature Science Foundation of China (No. 11204082) and Shanghai natural fund (No: 16ZR1410700).

**Conflicts of Interest:** The authors declare no conflict of interest.

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
