# Peer review of "Controllable Synthesis of Special Reed-Leaf-Like Carbon Nanostructures Using Copper Containing Catalytic Pyrolysis for High-Performance Field Emission"

_applsci, doi:10.3390/app9030440_

Round 1
Reviewer 1 Report
Article Review
The Controllable Synthesis of Special Reed-Leaf-Like Carbon Nanostructure Using Copper Containing Catalytic Pyrolysis for High-Performance Field Emission
The authors investigate catalytic growth of carbon nanostructures using Cu and Cu-Cr as catalytic growth materials. Catalyst morphology is varied as well as the gas used to clean the catalyst (this was a little ambiguous). The structures grown were then characterized via SEM and TEM imaging as well as Raman and field emission performance on several candidate morphologies. The best performing morphology was a unique blend of carbon nanotubes and what the authors term carbon nanoleaves (or read-leaf-like carbon nanostructures).
Overall, the presentation is poor and characterization is incomplete. Nevertheless, this article could be published if a number of issues can be addressed. The main problems fall into three main categories:
1) English/grammar
2) Proposed growth mechanism seems speculative and could have more experimental support
3) Elemental mapping (e.g. EDS and EELS) would put the characterization on much stronger footing
The English/grammar is poor and difficult to read. As a result, many sentences are somewhat ambiguous in meaning. A native English speaker needs to heavily edit the document before publication.
FE is not defined at first use in the abstract
Line 63: while it is appropriate to cite your previous work for more details, the authors should at least give a brief description of how the testing was performed here.
When discussing figure 1 in the text each panel should be briefly described so the reader understands what the figure is depicting. The details here are only found in the figure caption.
The scale bar text in figure 1 is too small.
The authors should consider providing a sentence or two describing why increasing catalyst film thickness results in increased particle size and roughness with citations to relevant studies.
The text associated with figure two (starting at line 75) does not describe the information from the figure caption. This should also appear in the main text.
Line 75: How are CNTs grown on the catalyst in an H2 atmosphere? Where does the carbon come from?
Line 78: The density of CNTs decreased significantly compared to what?
Figure 2 c and d: do the authors know the composition of the observed particles and “sub-micron nanofibers”. These are possibly carbon but it has not been demonstrated.
The substrate texture from the SEM images in figure 2 c and d do not appear to have similar morphology to the surfaces shown in figure 1. The texture should be visible in the SEM as well. Perhaps SEM images before and after growth would offer a clearer comparison.
Line 97: Authors mention “a series of works” but do not cite any articles there.
Line 100: TEM is not defined at first mention.
Figure 3b scale bar and text is too small. Figure 3 d has no scale bar.
The bamboo structure mentioned in line 101 should be more clearly shown in the figure. Consider a zoomed in overlay.
The authors suggest a growth mechanism (figure 4) for the carbon nanoleaf structures which is highly speculative. The growth mechanism involves distortion of the Cu particles into spindles. These spindles should be easily observed in TEM. Are they? In addition, the authors suggest that the introduction of nitrogen into the graphite sheet leads to distortion. The presence of nitrogen in these structures should be verified via EDX or EELS. These measurements would at least verify that the final structure is as described.
Author Response
Thank you for your useful comments and suggestions on the language and the structure of our manuscript. We have modified the manuscript accordingly, and the detailed corrections are listed in the document

Reviewer 2 Report
The paper describes in great details the characterization of the controlled synthesis of special reed-leaf-like carbon nanostructures using Cu containing catalytic pyrolysis. This leads to come improvement in the FE characteristics as shown in Fig,7 (reduction in threshold voltage and increase in FE current density). The increase in current density is not dramatic and Fig.7 would benefit from showing error bars associated to measurements on several samples. How reproducible are the FE measurements shown in Fig.7.
The text needs some improvement in the presentation. Some corrections were suggested
in an attached copy of the manuscript. Some sentences were rather obscure and difficult
to comment. The authors should rewrite these sentences.

Author Response

(The authors gave the same response as above.)

Reviewer 3 Report
The present manuscript deals with the synthesis of special reed-leaf-like carbon nanostructure using copper containing catalytic pyrolysis to study their field emission. Reviewer has carefully gone through the manuswcript. Manuscript is written neatly and well organised. The better FE properties of reed leaf-CNT composites is claimed which are superior CNT and CNL flowers, which is due to the floral clusters formed by inter-planting of CNT and CNL. it is suggested that FE stability of CNT and priority electron emission of defects in CNL play a vital role to improve the FE properties.Therefore, reviewer feels that this manuscript can be accepted in the present form.Author Response
Thank you for your useful comments and suggestions on the language and the structure of our manuscript. We have modified the manuscript accordingly, and the detailed corrections are listed below in the document

Round 2
Reviewer 1 Report
The majority of my concerns have been adequately addressed except for two.
1) The authors should include SEM images of the substrates before annealing for each subfigure in figure 2. The authors provided two such images in their response, but these were not included in the revised manuscript. This could be included in supplementary info but I suggest including it directly next to the other subfigures in figure 2. These “before” images help highlight the morphological changes that have occurred during annealing. I believe this will significantly improve the reader’s understanding when going through the figures.
2) Unfortunately, I do not consider the English/grammar to be significantly improved. I suggest the editors read the manuscript and judge for themselves. As this is not a specialized scientific problem perhaps the editors could work with the authors to improve the language. For example, the first two sentences read:
Special reed-leaf-like carbon nanostructure (CNL) formed of carbon nanotube (CNT) has been realized by using chemical vapor deposition (CVD) under the combined action of copper containing catalytic pyrolysis and ammonia (NH3) gas. The nucleation and growth mechanism of CNL based on the growth parameter are discussed.
But could be improved to:
Special reed-leaf-like carbon nanostructures (CNL) formed of carbon nanotubes (CNT) have been realized by using chemical vapor deposition (CVD) under the combined action of copper containing catalytic pyrolysis and ammonia (NH3) gas. The nucleation and growth mechanisms of CNLs based on the growth parameters are discussed.
I believe these details are specifically within the job description of an editor.
Author Response

(The authors gave the same response as above.)
